# Potential application of Aloe Vera-derived plant-based cell in powering wireless device for remote sensor activation

**Peng Lean Chong**[1]*, **Ajay Kumar Singh**[1], **Swee Leong Kok**[2]

**1** Centre for Communication System and IC Design (CSID), Faculty of Engineering and Technology, Multimedia University, Melaka, Malaysia, **2** Centre for Telecommunication Research & Innovation (CeTRI), Fakulti Kejuruteraan Elektronik dan Kejuruteraan Komputer (FKEKK), Universiti Teknikal Malaysia Melaka (UTeM), Melaka, Malaysia

\* plchong@mmu.edu.my

**Data Availability Statement:** All relevant data are within the manuscript only with attach figures.

**Funding:** This work was supported by MMU Mini Fund 2016/2017, MMUI/170005 to PLC and MMU R&D CAPITAL EXPENDITURE (CAPEX) 2018,

## Abstract

It is well proven that electrical energy can be harvested from the living plants which can be used as a potential renewable energy source for powering wireless devices in remote areas where replacing or recharging the battery is a difficult task. Therefore, harvesting electrical energy from living plants in remote areas such as in farms or forest areas can be an ideal source of energy as these areas are rich with living plants. The present paper proposes a design of a power management circuit that can harness, store and manage the electrical energy which is harvested from the leaves of *Aloe Barbadensis Miller* (Aloe Vera) plants to trigger a transmitter load to power a remote sensor. The power management circuit consists of two sections namely; an energy storage system that acts as an energy storage reservoir to store the energy harvested from the plants as well as a voltage regulation system which is used to boost and manage the energy in accordance to a load operation. The experimental results show that the electrical energy harvested from the Aloe Vera under a specific setup condition can produce an output of 3.49 V and 1.1 mA. The harvested energy is being channeled to the power management circuit which can boost the voltage to 10.9 V under no load condition. The harvested energy from the plants boosted by the power management circuit can turn ON the transmitter automatically to activate a temperature and humidity sensor to measure the environmental stimuli periodically with a $t_{on}$ of 1.22 seconds and $t_{off}$ of 0.46 seconds. This proves that this new source of energy combined with a power management circuit can be employed for powering the wireless sensor network for application in the Internet of Things (IoT).

## Introduction

Energy harvesting is a widely focused research area which is targeted to derive electrical energy from external sources which can be stored to power up small autonomous electronic devices. With recent technology advancement, where the power consumption of electronic devices had

MMUI / CAPEX180008 to PLC. The funders had no role in study design, data collection and analysis, decision to publish, or preparation of the manuscript.

**Competing interests:** The authors have declared that no competing interests exist.

been significantly reduced, numerous energy harvesting approaches have been introduced for supplying power to power-efficient electronic devices or wireless sensor nodes (WSNs) [1–2]. As the charges of the batteries depleted over time due to power consumption, the energy harvesting system is needed to recharge these batteries to sustain the functionality of the devices. Numerous techniques have been proposed to recharge the batteries such as photovoltaic cells, wind energy, kinetic energy, radio frequency, magnetic field and thermoelectric [3–13]. Although these renewable power sources have been tested to be able to activate the device, however, they normally face a problem in its scalability, cost of installation and applicability in a wide range of topography. Besides, these power sources may face the dilemma of their spasmodic availability throughout the day. It has been observed that extracting energy during the night is not possible in a photovoltaic cell and thermoelectric, which are dependent on the sun. In addition, a problem in the consistency of energy generation is faced by kinetic and wind energy where the wind is not abundant during certain seasons. Furthermore, radio frequency and magnetic field face a problem of completely absent or scarcity of energy sources in certain remote areas. Hence, this research paper introduces another type of energy harvesting technique. The proposed technique harvests the electrical energy from the living plants that can provide energy to a load throughout the day as long as the plants remain alive. This energy harvesting technique is feasible to be adapted in an area where living plants are abundantly available such as in farms or forest. It is shown by researchers that electrical energy can be tapped from a poplar tree (*Populus X Canadensis Moench*) using electrodes embedded into them [14]. In this research, the plant chosen is an *Aloe Barbadensis Miller* (Aloe Vera) which is from a succulent family of plants. Succulent plants are water-retaining plants, which can store water in their leaves, stems, and roots to survive in a dry environment. Due to this condition, the succulent family of plants that have a higher conductivity can generate a higher current [15]. Aloe Vera plants are chosen by the author compared to other succulent plants such as cacti because they are easily available in abundant in the wild in tropical and subtropical region such as in Malaysia where the experiment was conducted.

The electrical energy harvested from the living plant can be stored in a capacitor which can be used as a potential energy source to activate low power consumption devices such as a wireless transmitter paired to a sensor. However, a suitable power management circuit is needed to harvest, store and properly channel the energy to activate the transmitter to switch ON the sensor in a periodic manner. The importance of the power management circuit is shown in previous researches where electrical energy harvested from living plants was fed to a power conditioning circuit to enable powering of a load. Aloe Vera was being used as a specimen for energy harvesting and the electrical energy tapped measured at 0.945 V was fed to a simple external conditioning circuit consists of a transistor, resistor and an inductor coil which could light up a light emitting diode (LED) dimly [16]. In another experiment, electrical energy was harvested from a Bigleaf maple tree (*Acer Macrophyllum*) which was fed to two specialized nano-electronic ICs which consisted of a boost converter and a low frequency timer [17]. The electrical energy tapped from the tree was measured at 50 mV to 230 mV and 0.5 uA to 2.3 uA. The boost converter that was built via a 130 nm CMOS process could boost the voltage harvested from the tree to a higher voltage of approximately 1.1 V. Then, the low frequency timer build using a 90 nm CMOS process was operated based on the 1.1 V output from the boost converter. It was shown that the system could power the circuit in a nanoscale due to its minute energy. In an alternative experiment, electrical energy harvested from a Pachira tree (*Pachira Aquatic*) was used to power a wireless plant health monitoring system [18]. The electrical energy harvested from the tree was measured at 0.8 V and 3 uA. It was fed to a prototype circuit consisted of an intermittent power-gated supply circuit, storage capacitor and a DC-DC converter. The circuit could boost the electrical energy harvested from the plant from

**Fig 1. Anatomy of the Aloe Vera Plant.** In this figure, (a) shows the physical outlook of an Aloe Vera, (b) shows the cross-section of an Aloe Vera leaf, (c) shows the Aloe Vera rind layer, (d) shows the Aloe Vera latex layer and (e) shows the inner semi-solid fleshy gel of the Aloe Vera.

0.8 V to 2.0 V. The generated voltage was subsequently used to power a 300 MHz wireless transmitter. Hence, it is observed that a power management circuit which aims to manage the electrical energy tapped from a living plant must contain a boost converter that can step up the minute electrical energy produced by the plant to a higher level and an energy storage reservoir which can store the energy for further usage to power a load.

In the present paper, we have proposed a power management circuit, which can harvest the electrical energy from the Aloe Vera plants and converts the plants into a plant-based cell (PBC) to activate a remote sensor via a wireless transmission. The whole paper is organized as follows; section II explains the anatomy of the Aloe Vera plant, in brief, section III discusses the materials and methods used in the energy harvesting technique as well as in the design of the power management, section IV discusses the experimental results and finally section V concludes the paper.

## Anatomy of Aloe Vera plant

Aloe Vera plant is a type of water-retaining plant from the succulent species of plants, which can survive in an arid environment for a long period. It is a single stem plant growing at a height of approximately 50 cm to 100 cm with its large basal leaves growing in a spiral pattern around its stem at the center with thick fibrous root. The leaves are approximately one to two feet long and three to five cm wide depending on the size of the plants as shown in Fig 1(A). This plant is termed as "Lily of Desert" for its ability to store a high amount of water in its leaves. Its leaves are green, thick and fleshy with spiny edges at both sides. These fleshy leaves have pulps that can retain a high amount of water in the form of transparent gel. The cross-section of the Aloe Vera leaf (Fig 1(B)) shows that it consists of three layers, which are the Aloe rind, Aloe latex, and its inner juicy gel. The first layer is the Aloe Vera rind which is a green outer layer covering the leaf to protect its inner layers as shown in Fig 1(C). It has a waxy cover surface and it consists of cellulose materials with chlorophyll, which enables it to detect the light direction and generate photosynthesis. The second layer is the Aloe latex, which is a sap between the Aloe rind and the inner fleshy part as shown in Fig 1(D). The sap is a yellowish and bitter fluid that has an odor. The third layer is a clear inner semi-solid fleshy gel as shown in Fig 1(E) which is rich in nutrients and contains 99% water, and 1% of polysaccharides, glucomannans, lipids, amino acids, vitamins, sterols, minerals, and enzymes. This part is the largest portion of the leaf substance, which serves as the water storage reservoir for the plant. This part of the leaf also acts as an electrolyte in the electrochemistry process to generate electrical energy.

## Materials and methods

This section is divided into two subdivisions that cover first the material and method to investigate the specific setup condition to harvest electricity which is higher than 3 V and 1 mA

from the Aloe Vera plants and the second subsection discusses the design of the power management circuit.

## Materials and methods for optimum energy harvesting method

The analysis had been done on the succulent family of plant and non-succulent type family of the plant as energy sources, covering *Alstonia scholaris* (Pulai tree) and *Musa acuminata* (Banana tree) for the non-succulent plants as well as *Aloe barbadensis Miller* (Aloe Vera) for succulent plant [16]. It is found that the succulent plant produces a much higher voltage compares to a non-succulent plant. Thus, various researches have been performed to study Aloe Vera as a potential energy source that covers the measurement of its electrical impedance [19–20] and its electrochemistry as the origin of electrical energy when the plant is embedded with electrodes [21]. However, in these studies, the investigation on the specific setup condition to harvest electricity which is higher than 3 V and 1 mA from the Aloe Vera plants is not being explored. An in-depth investigation in the previous study [22] to harvest maximum electrical energy from the Aloe Vera plant using optimum experimental setups to charge the energy storage capacitor day and night had been performed. The charging rate of the capacitor is higher during the night due to the plants' respiration process. This is due to the cellular respiration process where, chemical energy in the glucose molecule generated via photosynthesis during the day is converted into form used by the plant for growth at night. The respiration process happens in two steps. The first step breaks the glucose material into two smaller molecule termed as pyruvate with energy release in the form of adenosine 5'-triphosphate (ATP). In the second step, the pyruvate molecules are rearranged and combined in a cyclic manner where carbon dioxide is produced and electrons are being extracted to be passed into an electron transport system which generates a higher number of ATP used in plant growth and reproduction. We have also observed that when a larger number of Cu-Zn electrode pairs arranged with the lesser distance between each of these electrodes, it can harvest a larger voltage and current from the plant. It is observed that by connecting the Aloe Vera leaves in series passing through the stem of the plant can increase the magnitude of the harvested voltage while the parallel connection among the leaves can increase the magnitude of the harvested current.

This section is intended to extend our previous research [22] to enable harvesting of voltage targeted to be higher than 3.0 V and current to be higher than 1.0 mA which, is suitable to be paired with a power management circuit to activate a specific load. The selected Aloe Barbadensis Miller plants are around 3 years old and are fully-grown plants. Aloe Vera plant will take approximately 3 to 4 years to grow from a pup into a full-grown plant where its leaves matured to its larger size measured 20 to 25 cm in length. The length of the leaves required is to be at least 20 cm in order to enable insertion of 8 electrode-pairs steadily into the leaves without causing any leaves to break into separate segments. Each of the plants selected here has a height of around 50 to 60 cm and a diameter from one tip of the leaf to another tip of the leaf is approximately 50 to 70 cm. The length of the Aloe Vera leaves is around 35 to 40 cm and the maximum width is 6 to 7 cm. The measured thickness is 2 to 2.5 cm. All the experimental setups are performed in an indoor laboratory environment with a room temperature of 25 to 26 degrees Celsius with relative humidity about 56% to 61%. The experimental setups are arranged next to a closed transparent window and the plants are subjected to the light intensity variation from the outdoor sunlight. A high precision Extech EX540 multimeter with data logging and wireless PC interface capability, with an accuracy of ±0.06% for voltage measurement range from 0.01 mV to 1000 V and the current measurement range from 0.01 μA to 20 A, is used to measure the voltage and current. Proper cleaning of the electrode pairs, used in the experiments, has been done by using sandpaper and alcohol to remove any contaminants

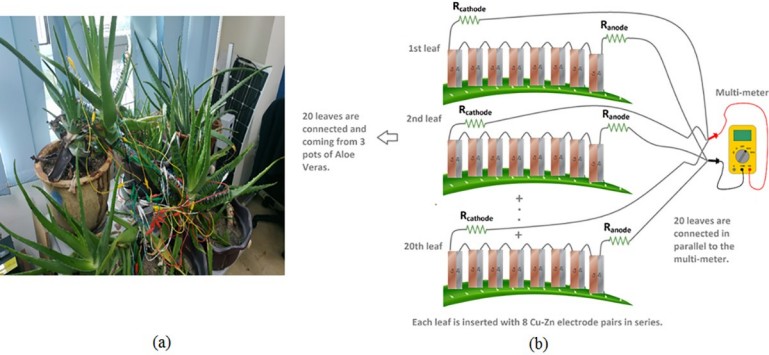

**Fig 2. The parallel connection among 20 leaves from 3 pots of Aloe Vera plants.**

before starting the experiment. All the materials and equipment used in this study are obtained from this academic institution, Multimedia University. The laboratory protocol is available at dx.doi.org/10.17504/protocols.io.2yngfve.

This research focuses on the investigation of the parallel connection between higher numbers of leaves among multiple Aloe Vera plants. Each leaf is embedded with a larger number of electrodes which are connected in series per leaf to boost the output voltage and current to a value of more than 3.0 V and 1 mA current in order to meet the power management circuit design criteria to operate the load. This setup (Fig 2) uses 8 electrode pairs per leaf compared to previous research [22]. The distance between each electrode is set at 1 cm. The same criterion is used for 20 individual leaves with 8 electrode pairs per leaf. This gives the total usage of 160 pairs of Cu-Zn electrodes. Then, all the 20 leaves are connected in parallel to increase the amount of current which can be harvested from multiple Aloe Vera plants. Three pots of Aloe Vera plants are used.

## Materials and methods for design of power management circuit

In most energy harvesting techniques, the harvester, which harvests energy from the external source, is not able to directly supply continuous power to the load due to its low instantaneous power compared to the power required for the wireless transmission [23–25]. However, as a wireless transmission often needs only intermittent data acquisition and periodically transmission of data to a receiver, thus a continuous supply of power to the transmitter is not required. The transmitter is only required to be powered up from time to time when enough amount of energy is needed for its operation. Therefore, a power management circuit is required to store and channelize sufficient energy needed to operate the load at an appropriate interval. The proposed power management circuit includes two sub-circuits; an energy storage system and a voltage regulation system as shown in Fig 3. The energy storage system acts as an energy

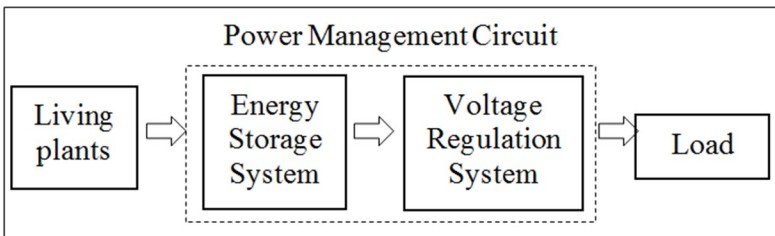

**Fig 3. Block diagram of the energy harvesting system.**

**Fig 4. Schematic of the proposed power management circuit connected to Aloe Vera.**

storage reservoir which is capable to accumulate an adequate amount of energy to be harvested from the Aloe Vera to achieve a certain threshold voltage before periodically activating the circuit. The voltage regulation system is used to manage the amount of voltage required to fit the operation of the load to function smoothly. Therefore, the design of a power management circuit is often based on the characteristic of an energy harvester. The proposed power management circuit aims to enable a specific interface, which can collect and manage electrical energy generated from the Aloe Vera plants with the limited current capability as well as provide the required output power to power a wireless transmitter to activate a temperature and humidity sensor. The apparatus which are used to make a precise measurement of the voltage and current in this research are Keithley DAQ6510 6½-digit data acquisition and logging multi-meter system and 3 units of EX540 multimeter with data logging and wireless PC interface.

The proposed power management circuit, shown in Fig 4, is designed to harvest enough electrical energy needed to operate a transmitter load.

The harvested energy from the 20 leaves of the Aloe Vera plants is accumulated periodically by the energy storage system, which consists of an input energy storage capacitor, $C_{in}$ and a voltage detector. First, the energy harvested from the plants is collected in $C_{in}$. The energy stored in $C_{in}$ will progressively increase the input voltage ($V_{Cin}$) across capacitor $C_{in}$ until sufficient energy has been accumulated in $C_{in}$ before the charges are released by the voltage detector. The voltage detector consistently monitors the maximum required voltage to be achieved by $C_{in}$. This voltage is set to be 3.0 V in this experiment. During the charging period of $C_{in}$ towards 3.0V, the voltage detector suspends the operation of the voltage regulation system and the transmitter load so that the voltage and energy harvested from the plants are fully stored in $C_{in}$. During this period, the wireless transmitter load remains inactive and electrically isolated from the plant power source. Once sufficient energy is accumulated in $C_{in}$ (more than 3.0V), the voltage detector will allow $C_{in}$ to discharge the stored electrical energy stored to the capacitor $C_1$. The charging and discharging process of $C_{in}$ is repeated which generates a signal of duty cycle, D, which is defined as:

$$D = \frac{t_{on}}{T} = \frac{t_{on}}{t_{on} + t_{off}} \tag{1}$$

where $t_{on}$ is the transmission time interval when the voltage regulation system is triggered and the wireless transmitter is activated to transmit a signal to activate a temperature and humidity sensor. On the other hand, $t_{off}$ is the time interval when the charging of the $C_{in}$ takes place and the wireless transmitter is deactivated so that the temperature and humidity sensors are OFF. The period $T$ is the repetition of the charging and discharging of the input storage capacitor $C_{in}$.

The continuous charging and discharging process of the capacitor $C_{in}$ causes a ripple voltage that influences the transfer of electrical energy from $C_{in}$ to the output. Thus, a rectifier consists of diode $D_1$ and $C_2$ is included in the design as a smoothing filter to reduce the ripple that exists in the signal from $C_{in}$. This enables the energy storage system to produce a stable pulsating DC signal to the whole circuit. The transfer of charge from $C_{in}$ to $C_2$ ignites the operation of the voltage regulation system which consists of a dc-dc self-oscillating boost converter. The dc-dc self-oscillating boost converter steps up the bulk of the energy from $C_2$ and transfers it to the output energy storage capacitor $C_{out}$. This enables voltage at the input capacitor to be boosted to a higher level at the output capacitor. The boost converter is designed to operate in a continuous conduction mode where the energy is transferred from $C_2$ to $C_{out}$ continuously without waiting for the voltage or current to drop zero. However, to enable such condition, the switch, SW, which is an n-channel MOSFET, must turn ON and OFF automatically before the output voltage or the current to drop to zero. It needs a gate threshold voltage, $V_{GSTH-SW}$ in between 2 V to 4 V to switch on the MOSFET automatically. Hence, the gate terminal of the MOSFET is connected to an LCR oscillating tank circuit which can act as a self-oscillator to provide sufficient oscillating signal to drive the MOSFET spontaneously. The oscillating tank circuit consists of an inductor, $L_f$ which is connected in series to a capacitor, $C_f$. Capacitor $C_f$ is connected in parallel with a resistor, $R_f$. The generated oscillating signal is capable to provide sufficient voltage to turn ON and OFF the MOSFET at the desired duty cycle.

During the continuous conduction mode, operation of the DC-DC boost converter is divided into two modes; Mode 1: when the MOSFET is turned ON and Mode 2: when the MOSFET is turned OFF. When the voltage level across $C_{in}$ is over 3V, it triggers the voltage detector to allow the electrical energy to pass through the LCR oscillating tank circuit. The LCR tank circuit generates a positive peak voltage to switch ON the MOSFET. This happens during Mode 1 and the charge from $C_{in}$ will flow through the inductor, L and the MOSFET drain and source as shown in Fig 5(A). This enables the charges to be accumulated in the inductor, L which is the key function to allow step-up of the voltage in the next mode. During Mode 2, the LCR oscillating tank generates a zero voltage to turn OFF the MOSFET. Hence the drain and source terminal of the MOSFET will be in open circuit condition. The current now will flow through the inductor L towards the diode, $D_2$ and to the output storage capacitor, $C_{out}$ as shown in Fig 5(B).

The stored energy in the inductor is now transferred to $C_{out}$ at the same time which boosts the output voltage, $V_{Cout}$ to be higher than the input voltage, $V_{Cin}$. The output voltage is expressed as:

$$V_{Cout} = \frac{V_{Cin}}{1 - D} \tag{2}$$

However, energy which is consumed by the load causes the inductor current to fall and the voltage level across $C_{in}$ to decrease below 3V which allows the voltage detector to isolate the

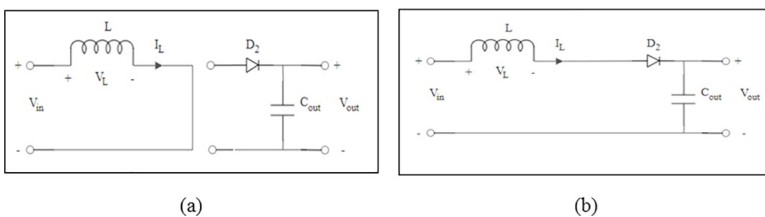

(a)                                                              (b)

**Fig 5.** Circuit diagram of boost converter during (a) $t_{on}$ (b) $t_{off}$.

voltage regulation system and the load again from the plant power source. This will restart the cycle of charging and discharging process of $C_{in}$. For the DC-DC converter to operate under the continuous conduction mode, the value of inductance of L must be selected according to the Eq (3) such that the current through inductor flows continuously and never falls to zero as according to the formula:

$$L_{min} = \frac{D(1-D)^2 R}{2f} \tag{3}$$

where $L_{min}$ is the minimum inductance of L, $R$ is output resistance of the load and $f$ is the switching frequency of the switch. Similarly, the output capacitance of $C_{out}$ must also be selected to achieve an output voltage with lesser fluctuation and low ripple to maintain a stable output voltage. The minimum output capacitance value $C_{min}$ can be expressed as:

$$C_{min} = \frac{D}{Rf\frac{\Delta V_o}{V_o}} \tag{4}$$

where $\frac{\Delta V_o}{V_o}$ is the output voltage ripple factor.

As the continuous transfer of charges takes place from $C_{in}$ to $C_{out}$, a stable supply of energy is fed to the load to support its operation. The load, which constitutes of a remote control circuit and a radio frequency (RF) transmitter module, has been selected such that it should function effectively when power is supplied via the stored energy in $C_{out}$. The remote controller selected is TX-2B, which is a CMOS LSI. It can operate between 1.0 V to 5.0 V with current less than 2 mA. The RF transmitter module is a 315 MHz module connected with an antenna which operates within 3 V to 12 V with current less than 2 mA. The TX-2B and transmitter operate in pairs for $t_{on}$ duration to transmit a signal to the receiver circuit when the charge is transferred from $C_{in}$ to $C_{out}$. During $t_{off}$, the $C_{in}$ will recharge to a level higher than 3.0 V to be ready for next cycle transmission. Hence, the cycle continues as it activates and deactivates the transmitter in sequence. Under this operating method, the transmitter will perform intermittent data acquisition and periodically transmission of the signal to the receiver circuit.

The receiver circuit constitutes a remote control circuit, RF receiver module, a micro-controller, and a temperature and humidity sensors as shown in Fig 6. The RF receiver module is a 315 MHz receiver module connected with an antenna. Once it receives a signal from the RF transmitter, it will send an activation signal to the encoding pin of the remote controller. The remote controller is an RX-2B module. Once received the signal, RX-2B will send a triggering signal to turn ON the base of the NPN transistor which controls the input power of the temperature and humidity sensors. The selected sensor is a DHT11 which operates between 3 V to 5.5 V. It can measure humidity in between 20–90% and the temperature from 0˚C -50˚C.

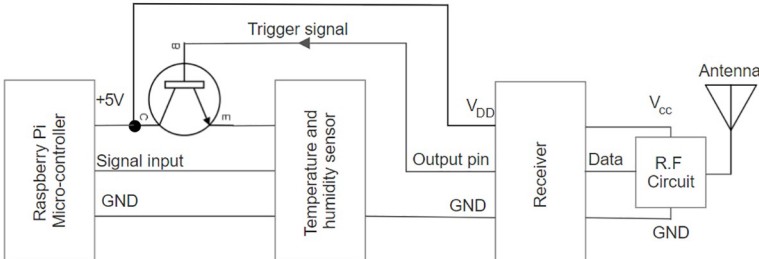

**Fig 6. Schematic of the receiver circuit with micro-controller connected to temperature and humidity sensor.**

Once the sensor is activated, the surrounding temperature and humidity will be measured and data is sent to the Raspberry Pi micro-controller for processing and to display the result at Thing Speak which is an open IoT web platform. The receiver circuit is fully supplied with power from the Raspberry Pi USB port.

## Results and discussion

This section of the paper is divided into two subdivisions. The first section covers the results and discussion for optimum energy harvesting method to tap an electrical energy higher than 3.0V and 1.0mA from the Aloe Vera plants and the second section covers the design of the power management circuit.

### Results and discussion for optimum energy harvesting method

The experimental results in terms of voltage and current, which can be harvested from 20 Aloe Vera leaves in parallel connection, are discussed in this section. As the number of parallel leaves is increased sequentially, the current increases. It is because each of the leaves which act as a cell contains some internal resistance. When more leaves are connected in parallel, the total internal resistance reduces and the admittance increases. This allows the increment of current as defined in Ohms law which states that when resistance decreases, the current will increase. The voltage is influenced by the number of Cu-Zn electrodes embedded per leaf. It is standardized to be 8 electrode pairs per leaf connected in series to achieve a consistency of harvesting voltage higher than 3.0 V from a leaf. Table 1 shows that the increment of number of electrode-pairs per leaf increases the amount of voltage generated.

For the parallel connection among 20 leaves, it is observed from Table 2 that the setup can generate a voltage ranging from approximately 3.454 V to 3.498 V when 8 pairs of Cu-Zn electrodes are embedded into each leaf. It is also observed that the higher number of leaves connected in parallel, the higher magnitude of current harvested from the system. The current can be harvested up to approximately 1.1 mA when all the 20 leaves are connected in parallel. As the current increases, the power harvested from the plants also increases. The maximum power to be harvested from 20 Aloe Vera leaves is 3.877 mW. The power management circuit is designed based on the input criterion, voltage larger than 3.0 V and current 1.1 mA, to manage the energy harvested from Aloe Vera plants to trigger the transmitter load.

### Results and discussion for design of power management circuit

There are several criterions to be considered in selecting a suitable input capacitor $C_{in}$ for the energy harvesting system. Its capacity should be large enough to accumulate sufficient energy to the desired level of input voltage $V_{Cin}$ before it is boosted by the self-oscillating boost

**Table 1. Voltage measured when varying numbers of electrode-pairs connected in series into a leaf.**

| Number of electrodes | Voltage measured, (V) |
|---|---|
| 1 | 0.970 |
| 2 | 1.455 |
| 3 | 1.860 |
| 4 | 2.181 |
| 5 | 2.515 |
| 6 | 2.912 |
| 7 | 3.210 |
| 8 | 3.450 |

Table 2. Voltage and current measured from 20 Aloe Vera leaves connected in parallel connection.

| The number of leaves connected in parallel connection. | Voltage (V) | Current (uA) | Power (uW) |
|---|---|---|---|
| 1 (first leaf) | 3.4670 | 107.07 | 371.21 |
| 2 | 3.4883 | 167.76 | 585.20 |
| 3 | 3.4882 | 224.82 | 784.22 |
| 4 | 3.4875 | 271.15 | 945.64 |
| 5 | 3.4858 | 335.44 | 1169.28 |
| 6 | 3.4978 | 385.08 | 1346.93 |
| 7 | 3.4747 | 422.22 | 1467.09 |
| 8 | 3.4850 | 524.18 | 1826.77 |
| 9 | 3.4710 | 558.40 | 1938.21 |
| 10 | 3.4867 | 613.90 | 2140.49 |
| 11 | 3.4798 | 686.60 | 2389.23 |
| 12 | 3.4652 | 720.90 | 2498.06 |
| 13 | 3.4545 | 781.18 | 2698.59 |
| 14 | 3.4978 | 823.58 | 2880.72 |
| 15 | 3.4876 | 856.75 | 2988.00 |
| 16 | 3.4897 | 890.81 | 3108.66 |
| 17 | 3.4864 | 920.96 | 3210.83 |
| 18 | 3.4805 | 957.17 | 3331.43 |
| 19 | 3.4990 | 981.05 | 3432.69 |
| 20 | 3.4981 | 1108.51 | 3877.68 |

converter and transferred to $C_{out}$. However, the capacity of $C_{in}$ should not be too large because it will take a longer charging duration to increase $V_{Cin}$ up to the threshold voltage of 3.0 V. If the capacity of $C_{in}$ is too small, it will not be able to store sufficient charge to power the load. Due to the above concern, the input capacitor $C_{in}$ selected is an aluminium electrolytic capacitor with a specification of 5.5 V, 0.22 F. On the other hand, the capacity of the output capacitor, $C_{out}$ is usually depended on the energy requirement of the load operation. It needs to store enough amount of energy to sustain a single operation of the load while at the same time do not causes an over lengthy duration to recharge it. This is to meet the periodical activation of the load from time to time at a reasonable period. In the experiment, a 10 nF capacitor is chosen to sustain the operation of the load. The duration needed to start the overall system for the first time by charging the $C_{in}$ to a threshold voltage of 3.0V represented as $t_{cold-start}$ is portrayed in Fig 7. The measurement is done by using the Extech EX540 multi-meter. It is observed that the input capacitor, $C_{in}$ required a $t_{cold-start}$ of approximately 460 seconds to charge from 0 V to the threshold voltage 3.0 V for the first time. During the $t_{cold-start}$ period, the voltage regulation system (which consists of the self-oscillating boost converter) and the load are isolated from the plant power source by the voltage detector. Once the threshold is achieved, the voltage fluctuated between 3.0 V to 3.1 V and now the voltage regulation system will be activated. The voltage detector selected here is a TC54VC3002ECB713 with a SOT-23A-3 surface mount package.

The rectifier circuit, consists of $D_1$ and $C_2$, is included in the power management circuit to reduce the ripple in the input. This ripple voltage influences the stability of the input voltage. The stability of the input voltage is crucial for enabling the operation of the voltage regulation system which affects the output voltage ($V_{Cout}$). A comparison of the voltage $V_{Cin}$ and $V_{Cout}$ for the conditions when the rectifier is excluded from the circuit versus when the rectifier is included in the circuit is shown in Fig 8.

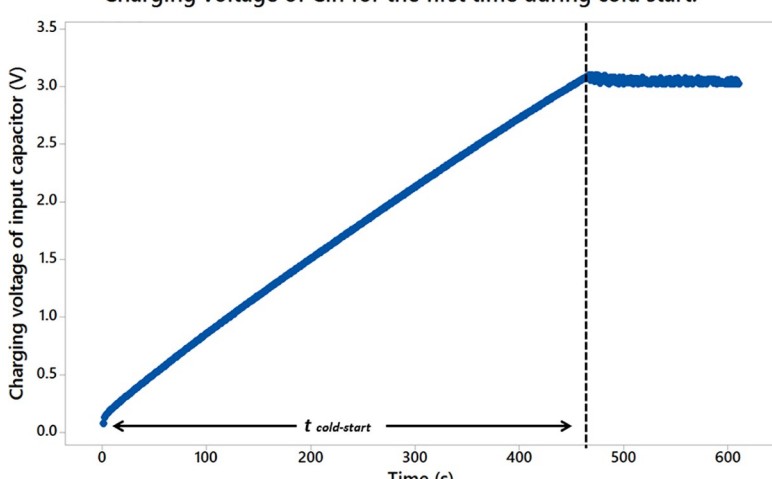

**Fig 7. Increment of V_Cin during the charging of the input capacitor for the first time.**

It is observed that large ripples occur in the $V_{Cin}$ and $V_{Cout}$ waveforms in Fig 8(A) when the rectifier circuit is excluded from the power management circuit. This causes the magnitude of the input and output voltages to continuously varying with the ripple and produces an unstable pulsating DC. The transmission time $t_{on}$ is approximately 250 ms and it is too short to enable any supply voltage to trigger the voltage regulation system. Meanwhile, the transmission off time $t_{off}$ is 470 ms, is higher than $t_{on}$. It results in a lower average voltage $V_{Cout}$ because the circuit stays in an off state longer time than in the active state. Hence, the average voltage $V_{Cout}$ is not enough to trigger the operation of the Tx-2B transmitter load which needs a minimum voltage of 1.0 V. The magnification of the waveform during $t_{on}$ is shown in Fig 8(B) with

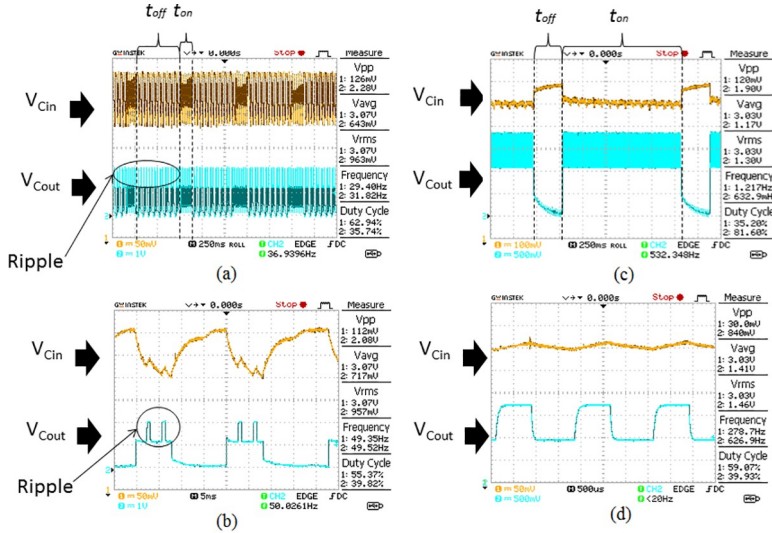

**Fig 8. Comparison of V_Cin and V_Cout when the rectifier is excluded from the circuit versus when the rectifier is included in the circuit.** (A) shows the waveform of $V_{Cin}$ and $V_{Cout}$ when the rectifier is excluded from the circuit. (B) shows the magnification of the waveform during the transmission time, $t_{on}$ when the rectifier is excluded from the circuit. (C) shows the waveform of $V_{Cin}$ and $V_{Cout}$ when the rectifier is included in the circuit. (D) shows the magnification of the waveform during the transmission time, $t_{on}$ when the rectifier is included in the circuit.

the average voltage of $V_{Cin}$ is at 3.07 V but $V_{Cout}$ is at 717 mV which does not meet the minimum voltage required to trigger the load. On the other hand, when the rectifier circuit is included in the circuit, the voltage $V_{Cin}$ and $V_{Cout}$ are observed to be more stable with fewer ripples as shown in Fig 8(C). This stable voltage across $C_{in}$ is fed as the input voltage to the voltage regulation system which generates a steady voltage at $C_{out}$ too. This output voltage is used to switch the transmitter load in a periodic manner. The transmission time, $t_{on}$ which is measured to be 1.38 seconds, is long enough to provide adequate time to generate a constant voltage to trigger the voltage regulation system. Meanwhile, $t_{off}$ is at 300 ms which is much lower compared to $t_{on}$. This enables the circuit to function in an active state longer than in the off state which increases the average output voltage. The higher value of $V_{Cout}$ provides sufficient voltage to switch ON the TX-RB transmitter load. The magnified waveform during $t_{on}$ in Fig 8(D) shows that there is no ripple observed in the waveform. The average voltage $V_{Cin}$ is at 3.03 V and $V_{Cout}$ is at 1.41 V which is sufficient to switch ON the transmitter load. The output voltage $V_{Cout,}$ measured when the rectifier is connected in the circuit, is much higher than the $V_{Cout}$ when the rectifier is excluded from the circuit. Hence, it is concluded that the rectifier circuit is needed in the design of the power management circuit to stabilize the output voltage and to enable a sufficient transmission time to trigger the load.

Next, the charge from $C_1$ is transferred to $C_2$ which acts as the input to the voltage regulation system (a self-oscillating boost converter). The boost converter is included with an LCR oscillating tank circuit which is used to switch on the electronic switch, SW automatically. The electronic switch must be chosen based on its voltage and current ratings which must be higher than the maximum input voltage and input current of the system. For the proposed system, the input voltage varies from 3.0 V to 3.1 V with approximately 1 mA current flow. Therefore, the selected electronic switch is an enhancement n-channel MOSFET IRF530N because its handling capability meets the specification of the proposed design. For the inductor L, the value selected must be higher than the calculated value based on Eq (3) which is 100 mH. For the selection of diode $D_1$ and $D_2$, the parameters concerning the diode reverse voltage rating, fast switching ability, low reverse recovery time, low forward voltage drop, blocking voltage during the off-state voltage stress and current handling capability are taken into consideration. Thus, both the diodes $D_1$ and $D_2$ are the ultra-fast recovery signal diode, 1N4148. For the selection of the output capacitor, $C_{out}$, the capacitance value should be higher than the minimum capacitance value calculated in Eq (4). In addition, consideration of the capacitor equivalent series resistance should be taken into account as it affects the capacitor efficiency in charging and discharging. Low equivalent series resistance is preferred for better performance and this can be achieved by connecting a few capacitors in parallel. The output capacitor selected here has a capacitance value of 10 uF. The LCR oscillating tank circuit is selected in such a way to provide an adequate gate to source voltage, $V_{GS-SW}$ to switch ON the MOSFET. The value of the components are $L_f$ = 22 uH, $R_f$ = 100 Ω and $C_f$ = 100 nF. With this combination, the LCR oscillating tank circuit can provide approximately 3.40 V to turn ON the MOSFET.

In order to evaluate the proposed self-oscillating boost converter, the proposed power management circuit is supplied with power from the Aloe Vera plant as shown in Fig 2 under no-load condition to harvest the maximum output voltage. From the experimental findings as shown in Fig 9(A), it is clear that the input voltage of 3.48 V can be boosted to 10.9 V at the output. The proposed design has a duty cycle of 68% as calculated from Eq (2). The switching characteristic of the MOSFET IRF530N which acts as the SW switch is shown in Fig 9(B). From Fig 9(B), it is observed that during time interval $t_{off}$, when the MOSFET is turned ON by gate voltage ($V_{GS-SW}$) of 3.40V (peak), the drain to source voltage ($V_{DS-SW}$) drops due to short-circuit between drain to source terminals which enables the drain current to flow through. During this period, the inductor L is being charged which blocks the current flow to the output

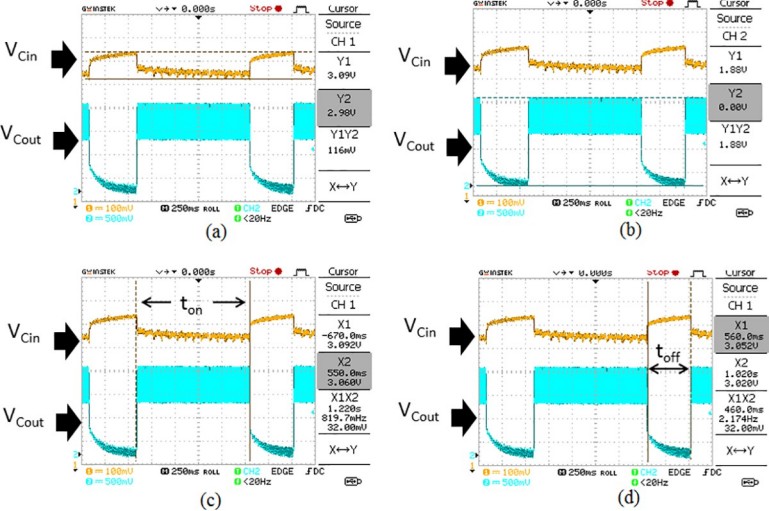

**Fig 9. Performance characteristic of the self-oscillating boost converter.** (a) shows the output voltage $V_{Cout}$ and $V_{Cin}$ of the power management circuit under no-load condition during the operation of the self-oscillating boost converter. (b) shows the switching characteristic of the SW switch during the operation of the self-oscillating boost converter under no-load condition.

capacitor, $C_{out}$. On the other hand, during time interval $t_{on}$, the $V_{GS-SW}$ is near to 0 V turns OFF the MOSFET. This causes the drain to source terminals to be in an open circuit condition and results in a peak voltage of ($V_{DS-SW}$) at 9.44 V. The previously-stored energy in inductor L as well as energy from input capacitor $C_{in}$ is transferred to $C_{out}$ at the same time. This condition steps up the input voltage from 3.48 V to 10.9 V at the output as shown in Fig 9(A). The experimental findings show that the self-oscillating boost converter is able to boost the input voltage to a higher magnitude at the output.

Next, the power management circuit is connected to the transmitter load. The Aloe Vera plants are supplying approximately 3.49 V and 1.1 mA to the whole system. The characteristic of the waveforms for $V_{Cin}$ and $V_{Cout}$ is shown in Fig 10. The energy harvested from the plants can charge the input capacitor up to 3.09 V which drops to 2.98 V as it discharges its energy to power the system as shown in Fig 10(A). The power management circuit transfers the charge to the output capacitor $C_{out}$ and this produces 1.88 V at the output which is sufficient to trigger the transmitter load as shown in Fig 10(B). The $V_{Cout}$ maintains at 1.88V for the duration, $t_{on}$ of 1.22 seconds before $C_{out}$ fully discharges to 0 Vas shown in Fig 10(C). Now, the input capacitor automatically recharges back to 3.09 V for the duration, 460 ms ($t_{off}$) to enable the

**Fig 10. The $V_{Cin}$ and $V_{Cout}$ waveforms when the power management circuit is connected to the transmitter load.** (a) indicates the changes in $V_{Cin}$. (b) indicates the variation of $V_{Cout}$. (c) portrays the active transmission time, $t_{on}$ when the transmitter load is triggered. (d) portrays inactive transmission time, $t_{off}$ when the transmitter load is not triggered.

**Fig 11. Temperature and humidity measured by the DHT 11 sensor.**

activation of the transmitter for the next cycle as shown in Fig 10(D). Hence, this process will continue in a sequence to trigger the transmitter load to send a signal to the receiver circuit and disable the transmitter load to enable recharging of the input capacitor by the Aloe Vera plants.

The received signal by the receiver circuit will activate the DHT 11 temperature and humidity sensors to measure the surrounding temperature and humidity in a periodic manner. Fig 11 shows the temperature and humidity measured by the DHT 11. The DHT11 is heated by using a hot gun from a Saike 852D++ soldering station from room temperature 26˚C up to 50˚C to test its functionality and humidity changes as it is activated by the signal from the transmitter. The transmitter is powered solely by using the power management system which harvests energy from the Aloe Vera plants. From the experiment (Fig 11) it is observed that the DHT 11 can measure accurately the change in temperature and humidity as it is heated from 26˚C up to 50˚C and then left to cool down progressively back to room temperature 26˚C. This result shows that the Aloe Vera plants combine with the power management system can act as a plant base cell (PBC) that can provide sufficient electrical energy to trigger a transmitter into activating a wireless sensor in a periodic manner.

Based on the result from this research, it is shown that the input voltage of 3.48 V harvested from the Aloe Vera leaves can be boosted to an output voltage of 10.9 V under a no-load condition by the power management circuit with a duty cycle of approximately 68%. Comparing the result with similar researches which targeted on energy harvesting from living plants as in [17] and [18], it shows that the power management circuit designed in this research shows an acceptable range of efficiency measured in duty cycle. As shown in [17], the input voltage of 50 mV to 230 mV harvested from a Bigleaf maple tree which is inserted with steel nails can be boosted to approximately 1.1 V via a boost converter built using a 130 nm CMOS process, with a duty cycle of 79% to 95%. However, the boost converter is built into an integrated circuit form which is fabricated specially to cater for ultra-low input voltage range in millivolt. On the other hand, in another experiment conducted to harvest electrical energy by inserting stainless steel electrode and iron nail into a Pachira tree as shown in [18], the energy harvested from the plant measured at 0.8V can be boosted to 2.0V by its DC-DC boost converter with a duty cycle of 60%. The DC-DC boost converter used is built based on discrete electronic components which is similar to the power management circuit shown in this research.

## Conclusion

In this paper, a plant base cell (PBC) has been proposed as a new electrical energy source to power low power consumption devices such as a transmitter. The PBC constitutes of a power management system that is connected to Cu-Zn electrode pairs which are embedded into the leaves of the Aloe Vera plants. The proposed power management system can perform a fully autonomous operation to harvest the electrical energy from the Aloe Vera plants to trigger a

transmitter load to send signal periodically to the temperature and humidity sensor. This has been confirmed by performing the experiment under a real-life condition. The designed power management circuit, which consists of an energy storage system and a voltage regulation system, can store the minute energy harvested from the Aloe Vera plants and boost them into sufficient energy to power a transmitter load. The transmitter load is proven to be in operation as it sends an intermittent signal to the receiver circuit to activate a remote sensor to measure the surrounding temperature and humidity. Thus, it is experimentally proven in this paper that Aloe Vera plants can be used as an energy source to provide electrical energy and its combination with the proposed power management circuit can act as a plant base cell. The idea of the proposed plant as a battery source can provide significant benefits in IoT application especially in remote areas or dense forest where replacing battery or recharging battery is impossible. The proposed cell can also be employed for precision farming and environmental monitoring where plants are available in abundant.

## Acknowledgments

We would like to thank Lim Wei Xiang and Han Aiman Rasyid for their valuable assistance in the fieldwork throughout the research. In addition, we also would like to thank Badrul bin Husin for providing the professional service to edit the manuscript in terms of language usage, spelling, and grammar.

## Author Contributions

**Conceptualization:** Peng Lean Chong.

**Data curation:** Peng Lean Chong.

**Formal analysis:** Peng Lean Chong.

**Funding acquisition:** Peng Lean Chong.

**Investigation:** Peng Lean Chong.

**Methodology:** Peng Lean Chong.

**Project administration:** Peng Lean Chong.

**Resources:** Peng Lean Chong.

**Software:** Peng Lean Chong.

**Supervision:** Peng Lean Chong, Ajay Kumar Singh, Swee Leong Kok.

**Validation:** Peng Lean Chong.

**Visualization:** Peng Lean Chong.

**Writing – original draft:** Peng Lean Chong.

**Writing – review & editing:** Peng Lean Chong, Ajay Kumar Singh, Swee Leong Kok.

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
