## [Decision Letter · Decision Letter 0]

18 Sep 2019

PONE-D-19-24896

Converting an Aloe Vera plant-based cell into powering a wireless device to activate a remote sensor

PLOS ONE

Dear Mr Chong,

Thank you for submitting your manuscript to PLOS ONE. After careful consideration, we feel that it has merit but does not fully meet PLOS ONE’s publication criteria as it currently stands. Therefore, we invite you to submit a revised version of the manuscript that addresses the points raised during the review process.

We would appreciate receiving your revised manuscript by Nov 02 2019 11:59PM. To enhance the reproducibility of your results, we recommend that if applicable you deposit your laboratory protocols in protocols.io, where a protocol can be assigned its own identifier (DOI) such that it can be cited independently in the future. For instructions see: http://journals.plos.org/plosone/s/submission-guidelines#loc-laboratory-protocols

We look forward to receiving your revised manuscript.

Kind regards,

Dai-Viet N. Vo, Ph.D.

Academic Editor

PLOS ONE

Journal Requirements:

Reviewers' comments:

Reviewer's Responses to Questions

**Comments to the Author**

1. Is the manuscript technically sound, and do the data support the conclusions?

Reviewer #1: Partly

Reviewer #2: Partly

2. Has the statistical analysis been performed appropriately and rigorously? 

Reviewer #1: No

Reviewer #2: Yes

3. Have the authors made all data underlying the findings in their manuscript fully available?

Reviewer #1: No

Reviewer #2: Yes

4. Is the manuscript presented in an intelligible fashion and written in standard English?

Reviewer #1: No

Reviewer #2: Yes

5. Review Comments to the Author

Reviewer #1: This paper presents the utilization of aloe vera as a potential energy source in low voltage electronic device. The overall idea of this paper is interesting, and the findings of this paper would add value to the total sum of human knowledge. However, this paper requires a significant improvement on the English. I suggest the author to consult a native English speaker to proofread their paper in order to better convey their messages and key findings to the reader. In addition, the paper should be formatted according to the Guide for Author before submission where some of the essential formatting standards should be included (line number, page number, double-spaced, etc). The current version is not reader friendly. Following are my comments for improvements:

Title:

I suggest the title can be revised as “Potential application of Aloe Vera-derived plant-based cell in powering wireless device for remote sensor activation”.

Abstract:

1. Please improve the English in terms of the grammar and sentence structure.

2. “Aloe Barbadensis Miller” Scientific name should be in italic form.

3. “….to store the harvested energy harvested from the plant….” Please re-write this sentence.

4. The values of important findings should be provided.

Introduction:

1. Please improve the English.

2. Define “WSNs” before it’s first mention.

3. “Populus X Canadensis Moench” Scientific name should be in italic form. Please correct the similar error as well.

4. Out of so many types succulent plant, why the author chooses Aloe Vera in the present study?

Anatomy of aloe vera plant:

1. The presentation of SI unit should be consistent where different forms were detected i.e. “cm” and “centimeter”. Please revise.

2. The contents of this section can be simply found on the internet and existing books. So, is this section necessary?

Materials and method:

1. “The charging rate of the capacitor is higher during the night due to the plants' respiration process.” Why? Please provide the explanation for this.

2. “The selected Aloe Barbadensis Miller plants are around 3 years old and” Can a younger plant be selected for such experiment (e.g. 1 year or 2 years old) consider the economic point of view and practicality?

Results and discussions:

1. “As the number of parallel leaves is increased sequentially, the current increases.” What could be the reason behind to this statement? The author should provide scientific explanation to this.

2. Did the author perform ANOVA analysis to their data obtained?

Reference:

The format of the journal should be consistent where some references are provided with DOI while some are not.

Reviewer #2: Comments:

1. Pg. 2: Power management design seems to be essential in this work, however it was not reviewed in detail in the introduction. The authors are advised to outline literature on the design of power management employed across literature in the introduction to highlight on the choice of this study’s technique.

2. Pg. 8: “The voltage is influenced by the number of Cu-Zn electrodes embedded per leaf. It is standardized to be 8 electrode pairs per leaf”.

This study outlines that its scope is to address the research gap for an optimum setup to harvest maximum energy from Aloe barbadensis Miller plant. However, the optimization variables on the materials and method selected are not clear in this study. In addition, it is also not clear on how the number of electrodes was concluded in this study to obtain the required voltage/current/power output as the previous study only mentioned up to 6 electrode pairs.

3. Pg. 9: The significance of this study is not obvious in terms of its power management design’s efficiency/duty cycle. The authors are advised to provide some comparison with existing power management designs as a benchmark.

6. PLOS authors have the option to publish the peer review history of their article (what does this mean?). If published, this will include your full peer review and any attached files.

Reviewer #1: No

Reviewer #2: No

---

## [Author Response · Author response to Decision Letter 0]

26 Nov 2019

Dear respected PLOS ONE Academic Editor and Reviewers;

Good day. Thank you for your kind comments which provide valuable insights to improve my current manuscript quality. Each of the comments given is well analyze and provided with necessary explanation, correction and response to support the points added into the manuscript. Hence, attach here is the comments and responses provided for your kind review and checking. 

Comments & Response for Reviewer 1:

Reviewer #1: This paper presents the utilization of aloe vera as a potential energy source in low voltage electronic device. The overall idea of this paper is interesting, and the findings of this paper would add value to the total sum of human knowledge. However, this paper requires a significant improvement on the English. I suggest the author to consult a native English speaker to proofread their paper in order to better convey their messages and key findings to the reader. In addition, the paper should be formatted according to the Guide for Author before submission where some of the essential formatting standards should be included (line number, page number, double-spaced, etc). The current version is not reader friendly. Following are my comments for improvements:

Response: The manuscript is send to a professional English lecturer, Mr Badrul Hisham to edit the manuscript in terms of language usage, spelling, and grammar. It is also been checked for its grammar usage via an online software, Grammarly. In addition, the manuscript is also formatted according to the Guide for Author to include the essential formatting standards such as line number, page number and double-spaced.

Comment 1:

Title:

I suggest the title can be revised as “Potential application of Aloe Vera-derived plant-based cell in powering wireless device for remote sensor activation”.

Response: The title is corrected as proposed. 

Comment 2: 

Abstract:

1. Please improve the English in terms of the grammar and sentence structure.

Response: The English in terms of the grammar and sentence structure is being checked, revised and corrected accordingly. 

2. “Aloe Barbadensis Miller” Scientific name should be in italic form.

Response: Corrected the term in italic form. 

3. “….to store the harvested energy harvested from the plant….” Please re-write this sentence.

Response: Sentence is corrected as “…to store the energy harvested from the plants…”.

4. The values of important findings should be provided.

Response: The value important findings are included in the abstract as shown below: “The experimental results show that the electrical energy harvested from the Aloe Vera under a specific setup condition can produce an output of 3.49V and 1.1mA. The harvested energy is being channeled to the power management circuit which can boost the voltage to 10.9V under no load condition. The harvested energy from the plants boosted by the power management circuit is capable to turn on the transmitter automatically to activate a temperature and humidity sensor to measure the environmental stimuli periodically with a ton of 1.22 seconds and toff of 0.46 seconds.”

Comment 3: 

Introduction:

1. Please improve the English.

Response: The English is edited, revised and improved by the author and proof read by a professional English lecturer

2. Define “WSNs” before it’s first mention.

Response: WSN is predefined before it is mention as : “…or wireless sensor nodes (WSNs) …”.

3. “Populus X Canadensis Moench” Scientific name should be in italic form. Please correct the similar error as well.

Response: The term is corrected to be in italic form.

4. Out of so many types succulent plant, why the author chooses Aloe Vera in the present study?

Response: The reason is explained in “Aloe Vera plants are chosen by the author compare to other succulent plants such as cacti as they are easily available in abundant in the wild in tropical and subtropical region such as in Malaysia where the experiment was conducted.”

Comment 4: 

Anatomy of aloe vera plant:

1. The presentation of SI unit should be consistent where different forms were detected i.e. “cm” and “centimeter”. Please revise.

Response: Revision done by using “cm” as SI unit only. 

2. The contents of this section can be simply found on the internet and existing books. So, is this section necessary?

Response: This section is included to enable the wide range of readers in PLOS ONE whom some might not familiar with Aloe Vera plants and its underlying anatomy which can store water and enable a higher conductivity in the plant for electric generation. 

Comment 5: 

Materials and method:

1. “The charging rate of the capacitor is higher during the night due to the plants' respiration process.” Why? Please provide the explanation for this.

Response: Explanation for the phenomenon is provided base on the plant respiration process as the following:

“The charging rate of the capacitor is higher during the night due to the plants' respiration process. This can be due to the cellular respiration process where, chemical energy in the glucose molecule generated via photosynthesis at the day is converted into form used by the plant for growth at night. The respiration process happens in two steps. The first step broke the glucose material into two smaller molecule termed as pyruvate with energy release in form of adenosine 5’-triphosphate (ATP). In the second step, the pyruvate molecules are rearranged and combined in a cyclic manner where carbon dioxide is produced and electrons are being extracted to be passed into an electron transport system which generates a higher number of ATP used in plant growth and reproduction.”

2. “The selected Aloe Barbadensis Miller plants are around 3 years old and” Can a younger plant be selected for such experiment (e.g. 1 year or 2 years old) consider the economic point of view and practicality?

Response: The selected Aloe Vera plants are around 3 years old and cannot be younger (at age 1-2 years old) because an Aloe Vera plant grown from a pup to a matured Aloe Vera at approximately 3 years only. During this stage, the Aloe Vera leaves will only reach a minimum length of 20cm which is needed in this experiments in order to insert 8 electrode-pairs ( which is 16 electrodes in total; 8 copper and 8 zinc electrodes with 1 cm distance between each). Some extra length is needed as the leaves of the plant is irregular in width with wider width near the stem and narrower width near the tip. Normally, electrode is not able to be inserted near the tip as it is narrow in size in comparable to width of electrode and when forcefully inserted near the tip will cause the leaves to broke into half. The explanation given in the thesis is shown below:

“The selected Aloe Barbadensis Miller plants are around 3 years old and are fully-grown plants. Aloe Vera plants will take approximately 3 to 4 years to grow from a pup into a full-grown plant where its leaves matured to its larger size measured 20 to 25 cm in length. The length of the leaves required is to be at least 20 cm in order to enable insertion of 8 electrode-pairs steadily into the leaves without causing any leaves to break into separate segments.”

Comment 6:

Results and discussions:

1. “As the number of parallel leaves is increased sequentially, the current increases.” What could be the reason behind to this statement? The author should provide scientific explanation to this.

Response: The explanation is provided in the manuscript as below:

“As the number of parallel leaves is increased sequentially, the current increases. This is due to each of the leaves which behave as a cell contains some internal resistance. When more leaves are connected in parallel, the total internal resistance reduces and the admittance increases, thus allowing the increment of current as defined in Ohms law which state that when resistance decreases, current will increase.”

2. Did the author perform ANOVA analysis to their data obtained?

Response: The one-way analysis of variance (ANOVA) is used to determine whether there are any statistically significant differences between the means of three or more independent (unrelated) groups. Hence, it is not applicable in the dataset in this journal as the dataset voltage (V), current (I) and power(P) are interdependent and related dataset by the formula P = I*V. 

Comment 7:

Reference:

The format of the journal should be consistent where some references are provided with DOI while some are not.

Response: Some articles do not have DOI as DOI is a fairly recent concept founded by the International DOI Foundation in 1998. Some publisher such as Elsevier started to use DOI on all their journals around 2003. However, some publishers do not implement DOI system in their article yet up to now. So, for journal number 17 ,18,19 and 20 which do not have ‘doi’, the citation is replace with ‘available from’ : (with link insert for the direct pdf file available online). 

Comments & Response for Reviewer 2:

Reviewer #2: Comments:

Comment 1:

1. Pg. 2: Power management design seems to be essential in this work, however it was not reviewed in detail in the introduction. The authors are advised to outline literature on the design of power management employed across literature in the introduction to highlight on the choice of this study’s technique.

Response: The outline of literature on the design of power management employed across literature in the introduction is added into the manuscript as below: 

“The electrical energy harvested from the living plant can be stored in a capacitor which can be used as a potential energy source to activate low power consumption devices such as a wireless transmitter paired to a sensor. However, a suitable power management circuit is needed to harvest, store and properly channelize the energy to activate the transmitter to switch ON the sensor in a periodic manner. The importance of the power management circuit is shown in previous researches where electrical energy harvested from living plants was fed to a power conditioning circuit to enable powering of a load. Aloe Vera was being used as a specimen for energy harvesting and the electrical energy tapped measured at 0.945V was fed to a simple external conditioning circuit consists of a transistor, resistor and an inductor coil which can light up a light emitting diode (LED) dimly [16]. In another experiment, electrical energy was harvested from a Bigleaf maple tree (Acer Macrophyllum) which was fed to two specialized nano-electronic ICs which consist of a boost converter and a low frequency timer [17]. The electrical energy tapped from the tree was measured at 50 mV to 230 mV and 0.5 uA to 2.3 uA. The boost converter was built on using a 130 nm CMOS process was able to boost the voltage harvested from the tree to a higher voltage of approximately 1.1 V. Then, the low frequency timer build using a 90 nm CMOS process was operated based on the 1.1V output from the boost converter. It is shown that the system was capable to power the circuit in a nanoscopic scale due to is minute energy. In an alternative experiment, electrical energy harvested from a Pachira tree (Pachira Aquatic) was used to power a wireless plant health monitoring system [18]. The electrical energy harvested from the tree was measured at 0.8 V and 3 uA. It was fed to a prototype circuit consists of an intermittent power-gated supply circuit, storage capacitor, DC-DC converter. The circuit was able to boost the electrical energy harvested from the plant from 0.8V to 2.0V. The generated voltage was subsequently used to power a 300 MHz wireless transmitter. Hence, it is observed that a power management circuit which aims to manage the electrical energy tapped from a living plant must contain a boost converter that can step up the minute electrical energy produced by the plant to a higher level and an energy storage reservoir which can store the energy for further usage to power a load.”

Comment 2:

2. Pg. 8: “The voltage is influenced by the number of Cu-Zn electrodes embedded per leaf. It is standardized to be 8 electrode pairs per leaf”.

This study outlines that its scope is to address the research gap for an optimum setup to harvest maximum energy from Aloe barbadensis Miller plant. However, the optimization variables on the materials and method selected are not clear in this study. In addition, it is also not clear on how the number of electrodes was concluded in this study to obtain the required voltage/current/power output as the previous study only mentioned up to 6 electrode pairs.

Response: Some correction is done on the statement given in the manuscript. The changes are listed here: 

This study outlines that its scope is to address the research gap for an optimum setup to harvest maximum energy from Aloe barbadensis Miller plant.

Answer: 

The study is corrected to be focusing on the specific setup to harvest electrical energy higher than 3.0 V and 1.0 mA from the Aloe Vera plant directly. The study to find optimum setup to harvested electrical energy is been focused and discussed in the author previous paper in [22]. Hence, it would be discussed briefly only in this current paper. The details are shown in Material and Method section: 

"However, in these studies, the investigation on the specific setup condition to harvest electrical energy measured to be higher than 3V and 1mA from the Aloe Vera plant is not being explored. We have performed an in-depth investigation in the previous study [22] to harvest maximum electrical energy from the Aloe Vera plant using optimum experimental setups to charge the energy storage capacitor day and night. “

“This section is intended to extend our previous research [22] to enable harvesting of voltage targeted to be higher than 3.0 V and current to be higher than 1.0 mA which, is suitable to be paired with a power management circuit to activate a specific load.”

The optimum setup to harvest higher amount of electrical energy is explained in the author’s previous journal in PLOS ONE cited [22] here in this manuscript. Hence, the optimum setup will not be explained in detail but briefly pointed out only here. The focus this time is more to explain the specific setup to obtain 3V and 1mA from the Aloe Vera via usage of 8 electrodes per leaf and by combining 20 leaves in parallel connection. It is portrayed in the statement below:

“This research focuses on the investigation of the parallel connection between higher numbers of leaves among multiple Aloe Vera plants. Each leaf is embedded with a larger number of electrodes which are connected in series per leaf to boost the output voltage and current to a value of more than 3.0 V and 1 mA current in order to meet the power management circuit design criteria to operate the load. This setup (Fig. 2) uses 8 electrode pairs per leaf compare to previous research [22]. The distance between each electrode is set at 1 cm. The same criterion is used for 20 individual leaves with 8 electrode pairs per leaf. This gives the total usage of 160 pairs of Cu-Zn electrodes. Then, all the 20 leaves are connected in parallel to increase the amount of current which can be harvested from multiple Aloe Vera plants. Three pots of Aloe Vera plants are used.”

For the results which shows that 8 electrodes are needed per leaf to enable generation of a voltage higher than 3.0V is stated in results and discussion as below: 

“The experimental results in terms of voltage and current, able to be harvested from 20 Aloe Vera leaves in parallel connection, is discussed in this section. As the number of parallel leaves is increased sequentially, the current increases. This is due to each of the leaves which behave as a cell contains some internal resistance. When more leaves are connected in parallel, the total internal resistance reduces and the admittance increases, thus allowing the increment of current as defined in Ohms law which state that when resistance decreases, current will increase. The voltage is influenced by the number of Cu-Zn electrodes embedded per leaf. It is standardized to be 8 electrode pairs per leaf connected in series to achieve a consistency of harvesting voltage higher than 3.0V from a leaf. Table 1 shows that the increment of number of electrode-pairs per leaf increases the amount of voltage generated.”

Table 1. Voltage measured when varying numbers of electrode-pairs connected in series into a leaf.

Number of electrodes Voltage measured, (V)

1 0.970

2 1.455

3 1.860

4 2.181

5 2.515

6 2.912

7 3.210

8 3.450

“For the parallel connection between 20 leaves, it is observed from Table 2 that the setup can generate a voltage ranging from approximately 3.454 V to 3.498 V when 8 pairs of Cu-Zn electrodes are embedded into each leaf. It is also observed that the higher number of leaves connected in parallel, the higher magnitude of current harvested from the system. The current can be harvested up to approximately 1.1 mA when all the 20 leaves are connected in parallel. As the current increases, the power harvested from the plants also increases. The maximum power to be harvested from 20 Aloe Vera leaves is 3.877 mW. The power management circuit is designed based on the input criterion, voltage larger than 3.0 V and current 1.1 mA, to manage the energy harvested from Aloe Vera plants to trigger the transmitter load.”

Table 2. Voltage and current measured from 20 Aloe Vera leaves connected in parallel connection.

The number of leaves connected in parallel connection. Voltage (V) Current (uA) Power (uW)

1 (first leaf) 3.4670 107.07 371.21

2 3.4883 167.76 585.20

3 3.4882 224.82 784.22

4 3.4875 271.15 945.64

5 3.4858 335.44 1169.28

6 3.4978 385.08 1346.93

7 3.4747 422.22 1467.09

8 3.4850 524.18 1826.77

9 3.4710 558.40 1938.21

10 3.4867 613.90 2140.49

11 3.4798 686.60 2389.23

12 3.4652 720.90 2498.06

13 3.4545 781.18 2698.59

14 3.4978 823.58 2880.72

15 3.4876 856.75 2988.00

16 3.4897 890.81 3108.66

17 3.4864 920.96 3210.83

18 3.4805 957.17 3331.43

19 3.4990 981.05 3432.69

20 3.4981 1108.51 3877.68

Comment 3:

3. Pg. 9: The significance of this study is not obvious in terms of its power management design’s efficiency/duty cycle. The authors are advised to provide some comparison with existing power management designs as a benchmark.

Response: The comparison of the power management circuit shown in this research with existing power management design in other researches aim for boosting the electrical energy harvested from living plants is shown in Result and Discussion section as shown below:

“Based on the result from this research, it is shown that the input voltage of 3.48 V harvested from the Aloe Vera leaves can be boosted to an output voltage of 10.9 V under a no-load condition by the power management circuit with a duty cycle of approximately 68% as stated earlier. Comparing the result with similar researches which targeted on energy harvesting from living plants as in [17] and [18], it shows that the power management circuit designed in this research shows an acceptable range of efficiency measured in duty cycle. As shown in [17], the input voltage of 50 mV to 230 mV harvested from a Bigleaf maple tree which is inserted with steel nails can be boosted to approximately 1.1 V via a boost converter built using a 130 nm CMOS process, with a duty cycle of 79% to 95%. However, the boost converter is built into an integrated circuit form which is fabricated specially to cater for ultra-low input voltage range in millivolt. On the other hand, in another experiment conducted to harvest electrical energy by inserting stainless steel electrode and iron nail into a Pachira tree as shown in [18], the energy harvested from the plant measured at 0.8V can be boosted to 2.0V by its DC-DC boost converter with a duty cycle of 60%. The DC-DC boost converter used is built based on discrete electronic components which is similar to the power management circuit shown in this research.”

---

## [Decision Letter · Decision Letter 1]

9 Dec 2019

PONE-D-19-24896R1

Potential application of Aloe Vera-derived plant-based cell in powering wireless device for remote sensor activation.

PLOS ONE

Dear Mr Chong,

Thank you for submitting your manuscript to PLOS ONE. After careful consideration, we feel that it has merit but does not fully meet PLOS ONE’s publication criteria as it currently stands. Therefore, we invite you to submit a revised version of the manuscript that addresses the points raised during the review process.

We would appreciate receiving your revised manuscript by Jan 23 2020 11:59PM. To enhance the reproducibility of your results, we recommend that if applicable you deposit your laboratory protocols in protocols.io, where a protocol can be assigned its own identifier (DOI) such that it can be cited independently in the future. For instructions see: http://journals.plos.org/plosone/s/submission-guidelines#loc-laboratory-protocols

We look forward to receiving your revised manuscript.

Kind regards,

Dai-Viet N. Vo, Ph.D.

Academic Editor

PLOS ONE

Additional Editor Comments (if provided):

Professional proofread should be conducted for the whole manuscript again.

Reviewers' comments:

Reviewer's Responses to Questions

**Comments to the Author**

1. If the authors have adequately addressed your comments raised in a previous round of review and you feel that this manuscript is now acceptable for publication, you may indicate that here to bypass the “Comments to the Author” section, enter your conflict of interest statement in the “Confidential to Editor” section, and submit your "Accept" recommendation.

Reviewer #1: All comments have been addressed

Reviewer #2: All comments have been addressed

2. Is the manuscript technically sound, and do the data support the conclusions?

Reviewer #1: Yes

Reviewer #2: Yes

3. Has the statistical analysis been performed appropriately and rigorously? 

Reviewer #1: Yes

Reviewer #2: Yes

4. Have the authors made all data underlying the findings in their manuscript fully available?

Reviewer #1: Yes

Reviewer #2: Yes

5. Is the manuscript presented in an intelligible fashion and written in standard English?

Reviewer #1: Yes

Reviewer #2: No

6. Review Comments to the Author

Reviewer #1: The author had carefully addressed to all of my comments. I recommend an acceptance to its current form.

Reviewer #2: The comments outlined have been addressed clearly. However, the reviewer recommends the manuscript to undergo some minor revisions to address the gramatical errors. Below are few of the examples for the authors’ reference:

1. Pg. 3: "The boost converter that was built on using a 130 nm CMOS process could boost the voltage harvested..."

2. Pg. 3: "It is shown that the system could power the circuit in a nanoscopic scale due to is minute energy."

3. Pg. 4 and 5: "...harvest electrical energy measured to be higher than 3 V and 1 mA from the Aloe Vera plant..."

4. Page 11: “This is due to each of the leaves which behave as a cell contains some internal resistance.”

5. Page 11: “When more leaves are connected in parallel, the total internal resistance reduces and the admittance increases, thus allowing the increment of current as defined in Ohms law which state that when resistance decreases, current will increase.”

7. PLOS authors have the option to publish the peer review history of their article (what does this mean?). If published, this will include your full peer review and any attached files.

Reviewer #1: No

Reviewer #2: No

---

## [Author Response · Author response to Decision Letter 1]

9 Dec 2019

Rebuttal Letter: Response to Reviewers PLOS ONE

Dear respected PLOS ONE Academic Editor and Reviewers;

Good day. Thank you for your kind comments which provide valuable insights to improve my current manuscript quality. Each of the comments given is well analyze and provided with necessary explanation, correction and response to support the points added into the manuscript. Hence, attach here is the comments and responses provided for your kind review and checking.

Comments & Response for Reviewer 1:

Reviewer #1: The author had carefully addressed to all of my comments. I recommend an acceptance to its current form.

Response: No correction needed.

Comments & Response for Reviewer 1:

Reviewer #2: The comments outlined have been addressed clearly. However, the reviewer recommends the manuscript to undergo some minor revisions to address the grammatical errors. Below are few of the examples for the authors’ reference:

1. Pg. 3: "The boost converter that was built on using a 130 nm CMOS process could boost the voltage harvested..."

Response: The sentence is corrected to be:

“The boost converter that was built via a 130 nm CMOS process could boost the voltage harvested…”

2. Pg. 3: "It is shown that the system could power the circuit in a nanoscopic scale due to is minute energy."

Response: The sentence is corrected to be:

“It was shown that the system could power the circuit in a nanoscale due to its minute energy.”

3. Pg. 4 and 5: "...harvest electrical energy measured to be higher than 3 V and 1 mA from the Aloe Vera plant..."

Response: The sentence is corrected to be:

“…harvest electricity which is higher than 3 V and 1 mA from the Aloe Vera plants…”

4. Page 11: “This is due to each of the leaves which behave as a cell contains some internal resistance.”

Response: The sentence is corrected to be:

“It is because each of the leaves which act as a cell contains some internal resistance.”

5. Page 11: “When more leaves are connected in parallel, the total internal resistance reduces and the admittance increases, thus allowing the increment of current as defined in Ohms law which state that when resistance decreases, current will increase.”

Response: The sentence is corrected to be:

“When more leaves are connected in parallel, the total internal resistance reduces and the admittance increases. This allows the increment of current as defined in Ohms law which states that when resistance decreases, the current will increase.”

---

## [Editor Report · Decision Letter 2]

13 Dec 2019

Potential application of Aloe Vera-derived plant-based cell in powering wireless device for remote sensor activation.

PONE-D-19-24896R2

Dear Dr. Chong,

We are pleased to inform you that your manuscript has been judged scientifically suitable for publication and will be formally accepted for publication once it complies with all outstanding technical requirements.

With kind regards,

Dai-Viet N. Vo, Ph.D.

Academic Editor

PLOS ONE

Additional Editor Comments (optional):

The paper has been properly revised. Thus, it could be considered for publication.
---

## [Editor Report · Acceptance letter]

18 Dec 2019

PONE-D-19-24896R2 

Potential application of Aloe Vera-derived plant-based cell in powering wireless device for remote sensor activation. 

Dear Dr. Chong:

I am pleased to inform you that your manuscript has been deemed suitable for publication in PLOS ONE. Congratulations! Your manuscript is now with our production department. 

With kind regards,

on behalf of

Dr. Dai-Viet N. Vo 

Academic Editor

PLOS ONE